# Unsupervised alignment reveals structural commonalities and differences in neural representations of natural scenes across individuals and brain areas

**Ken Takeda**[1][†] **, Kota Abe**[1][†] **, Jun Kitazono**[2] **& Masafumi Oizumi**[1]
[1]Graduate School of Arts and Science, The University of Tokyo, Tokyo, Japan
[2]Graduate School of Data Science, Yokohama City University, Kanagawa, Japan
`{tkkentakeda1248, abe-sim, c-oizumi}@g.ecc.u-tokyo.ac.jp`
`kitazono.jun.ig@yokohama-cu.ac.jp`

## Abstract

Neuroscience research has extensively explored the commonality of neural representations of sensory stimuli across individuals to uncover universal neural mechanisms for encoding sensory information. To compare neural representations across different brains, Representational Similarity Analysis (RSA) has been used, which focuses on the similarity structures of neural representations for different stimuli. Despite the broad applicability and utility of RSA, one limitation is that its conventional framework assumes that neural representations of particular stimuli correspond directly to those of the same stimuli in different brains. This assumption excludes the possibility that neural representations correspond differently and limits the exploration of finer structural similarities. To overcome this limitation, we propose to use an unsupervised alignment framework based on Gromov-Wasserstein Optimal Transport (GWOT) to compare similarity structures without presupposing stimulus correspondences. This method allows for the identification of optimal correspondence between neural representations of stimuli based solely on internal neural representation relationships, providing a more detailed comparison of neural similarity structures across individuals. We applied this unsupervised alignment to investigate the commonality of representational similarity structures of natural scenes, using large datasets of Neuropixels recordings in mice and fMRI recordings in humans. We found that the similarity structure of neural representations in the same visual cortical areas can be well aligned across individuals in an unsupervised manner in both mice and humans. On the other hand, we found that the degree of the alignment across different brain areas cannot be fully explained by proximity in the visual processing hierarchy alone, while we also found some reasonable alignment results, such that the similarity structures of higher-order visual areas can be well aligned with each other, but not with lower-order visual areas. We expect that our unsupervised approach will be useful for revealing more detailed structural commonalities or differences that may not be captured by the conventional supervised approach.

## 1 Introduction

In the field of neuroscience, researchers have long investigated the commonality of neural representations of sensory stimuli across individuals in an attempt to find the universal neural mechanisms for encoding sensory information. It is typically assumed that sensory information is represented as the population activity of neurons or brain areas (Haxby et al., 2001; Hasson et al., 2004). The difficulty in comparing the neural representations across different brains is that there are no correspondences between neurons in different brains (Fig. 1a).

One way to compare neural representations without correspondences between neurons is to focus on the representational similarity structures, known as Representational Similarity Analysis (RSA)

---

[†]These authors contributed equally to this work

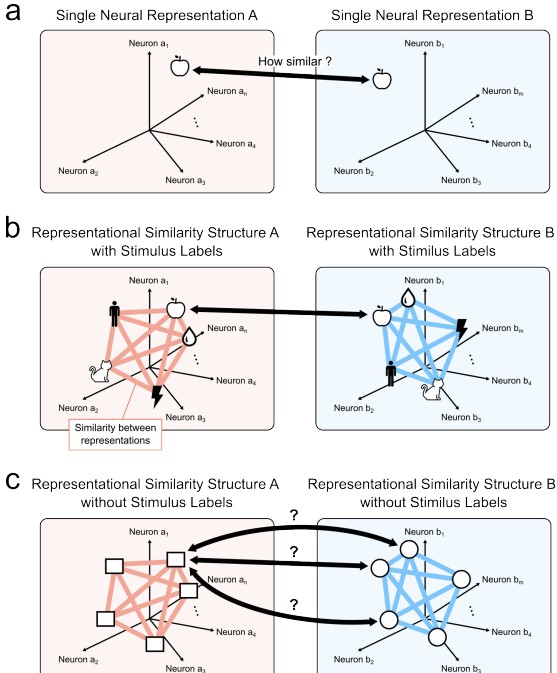

Figure 1: **Comparison of neural representations across different individuals**. (a) Comparison of single neural representations. Direct comparison of single neural representations across different individuals is challenging due to the lack of correspondence between neurons. (b) Supervised alignment: By assuming a correspondence among stimulus labels and conducting comparisons of "representational similarity structures", we enable quantitative comparisons of neural representations across different individuals. (c) Unsupervised alignment: In the situation where neural representations to different stimuli may correspond across individuals, the estimation of correspondence between neural representations must rely solely on each similarity structure, without the use of stimulus labels.

(Kriegeskorte et al., 2008a; Kriegeskorte & Kievit, 2013). RSA circumvents the need for direct neuron-to-neuron correspondences by focusing on the similarity structures within the neural representations of different stimuli (Fig. 1b). This method considers the representational similarity structures—the similarities or dissimilarities between the neural responses to different stimuli within each brain. By evaluating the similarity of these structures, RSA allows for the comparison of neural representations across brains, even in the absence of direct neuronal correspondences. Furthermore, RSA extends its utility beyond neural responses, enabling the comparison across different modalities, such as behavior or computational models, thereby offering a versatile framework for understanding the commonality or differences in neural representations between different systems (Cichy et al., 2019; Khaligh-Razavi & Kriegeskorte, 2014). By comparing the similarity structure of neural representations, this approach has suggested the presence of structural commonalities in neural representations among individuals in both humans and animals (Deitch et al., 2021; Charest et al., 2014; Kriegeskorte et al., 2008b; Raizada & Connolly, 2012; Shinkareva et al., 2012).

Despite its widespread application and usefulness, the conventional RSA framework is not without limitations. The framework typically assumes that there are direct correspondences between the neural representations of the same stimuli in different brains, which we call supervised comparison. This assumption is satisfactory for exploring gross-level similarities between neural representations, but falls short when exploring finer-level structural similarities. For example, this supervised framework inherently cannot answer the question of whether or not the neural responses to the same particular stimuli are represented in the same way in different brains, i.e., the validity of the correspondence assumption itself. In principle, it is possible that the neural response to one particular stimulus, such as an apple, may not always correspond directly to the same stimulus in another brain, but may be more closely mapped to the response to another stimulus, such as an orange, or may be mapped on a coarser group-to-group level, such as fruit-to-fruit, rather than on a precise one-to-one (apple-to-apple) level.

To address these limitations and to further extend the framework of representational similarity analysis even when there are no given correspondences between stimuli, we propose an unsupervised alignment framework for comparing similarity structures of neural representations without presupposing specific stimulus correspondences. The unsupervised alignment framework enables us to identify the optimal correspondence based solely on the internal relationship of neural representations, without presupposing any specific correspondence relationship (Fig. 1c), providing a more nuanced understanding of how neural similarity structures are shared across different individuals. For the unsupervised alignment method, we use Gromov-Wasserstein Optimal Transport (GWOT) for unsupervised alignment (Mémoli, 2011), a method that has been successfully applied in various

fields (Alvarez-Melis & Jaakkola, 2018; Demetci et al., 2022), including neuroscience (Kawakita et al., 2023a;b; Thual et al., 2022).

Our research leverages this unsupervised alignment framework based on GWOT to explore the commonality of representational similarity structures of natural scenes across different individuals. Specifically, we investigated this using large-scale open datasets of Neuropixels recordings in mice and fMRI recordings in humans. For mice, we used electrophysiological data from the Allen Brain Observatory (Siegle et al., 2021), collected using Neuropixels probes. For humans, we used the Natural Scenes Dataset(NSD)(Allen et al., 2022). These datasets are suitable for our research purposes because they contain a large number of natural scenes stimuli, allowing us to investigate the rich and complex similarity structures of neural representations of natural scenes. Furthermore, these datasets provide high-quality neural activity measurements across many individuals, enabling statistically reliable analysis. Using these ideal resources, we first investigated whether the similarity structures of natural scenes from the same brain regions can be aligned in an unsupervised manner across different individuals for both mice and humans. In addition, we also investigated whether the similarity of representations can be aligned across different brain regions. This approach opens new avenues for revealing more nuanced structural similarities or differences in neural representations.

## 2 RESULTS

In the following analysis, we performed an unsupervised alignment of neural representations. See the Methods A.1 for details on the method.

### 2.1 UNSUPERVISED ALIGNMENT OF MOUSE NEURAL REPRESENTATIONS

We first performed unsupervised alignment of neural representations in mice using the Neuropixels dataset (Siegle et al., 2021), to investigate whether neural representations of natural scenes can be aligned across different mice (see Methods A.2 for the data description). We analyzed the neural activity in 6 areas of the visual cortex (VISp, VISrl, VISl, VISal, VISpm, VISam), 1 area of the thalamus (LGd), and 1 area of the hippocampus (CA1). In the following, we first performed an unsupervised alignment between the same areas of different individual mice to investigate whether the representational similarity structures of the anatomically identical areas could be aligned across different mice. Then, we performed an unsupervised alignment between the different areas to investigate the commonalities of the similarity structures between the different areas.

In this study, we performed an unsupervised alignment between two pseudo-mice (see Methods A.3). We repeated the analysis 10 times, changing the division of the mice to construct different pairs of pseudo-mice each time. This approach was employed to examine the influence of mouse selection on the group-averaged representational similarity structures.

**Representational similarity structures in each brain area** To perform unsupervised alignment, we estimated the representational similarity structures of 118 natural scene stimuli for each brain area (see Methods A.3). In Fig. 2a1, we show a specific example of the dissimilarity matrices of a pair of pseudo-mice for the 118 stimuli in each area. We quantified the dissimilarity by the cosine distance between the trial-averaged normalized spike counts of two stimuli (see Methods A.3). As can be seen in Fig. 2a1, the dissimilarity matrices of two pseudo-mice are highly similar for the visual cortical areas (VISp, VISrl, VISl, VISal, VISpm, VISam). In these areas, the correlations between the dissimilarity matrices are about 0.6–0.9 (Fig. 2b). Compared to the visual cortical areas, the correlations in the thalamus (LGd) are lower, about 0.4 (Fig. 2b). In contrast to these areas in the visual system, the correlations in the hippocampus (CA1) are close to 0, indicating that there are no common structures in CA1.

**Unsupervised alignment between the same brain areas** We investigated whether there are sufficient structural similarities to allow unsupervised alignment of the neural representations within the same brain area across animals. To this end, we performed unsupervised alignment of the dissimilarity matrices in the same area across different pseudo-mice (Fig. 2a). For each brain area, we performed GWOT with 100 trials on different $\epsilon$ values. The points in Fig. 2a2 correspond to the estimated GWD (Eq. 1) on each optimization trial. We selected the result of the trial with the lowest estimated GWD as the optimal solution (shown in the blue circle in Fig. 2a2).

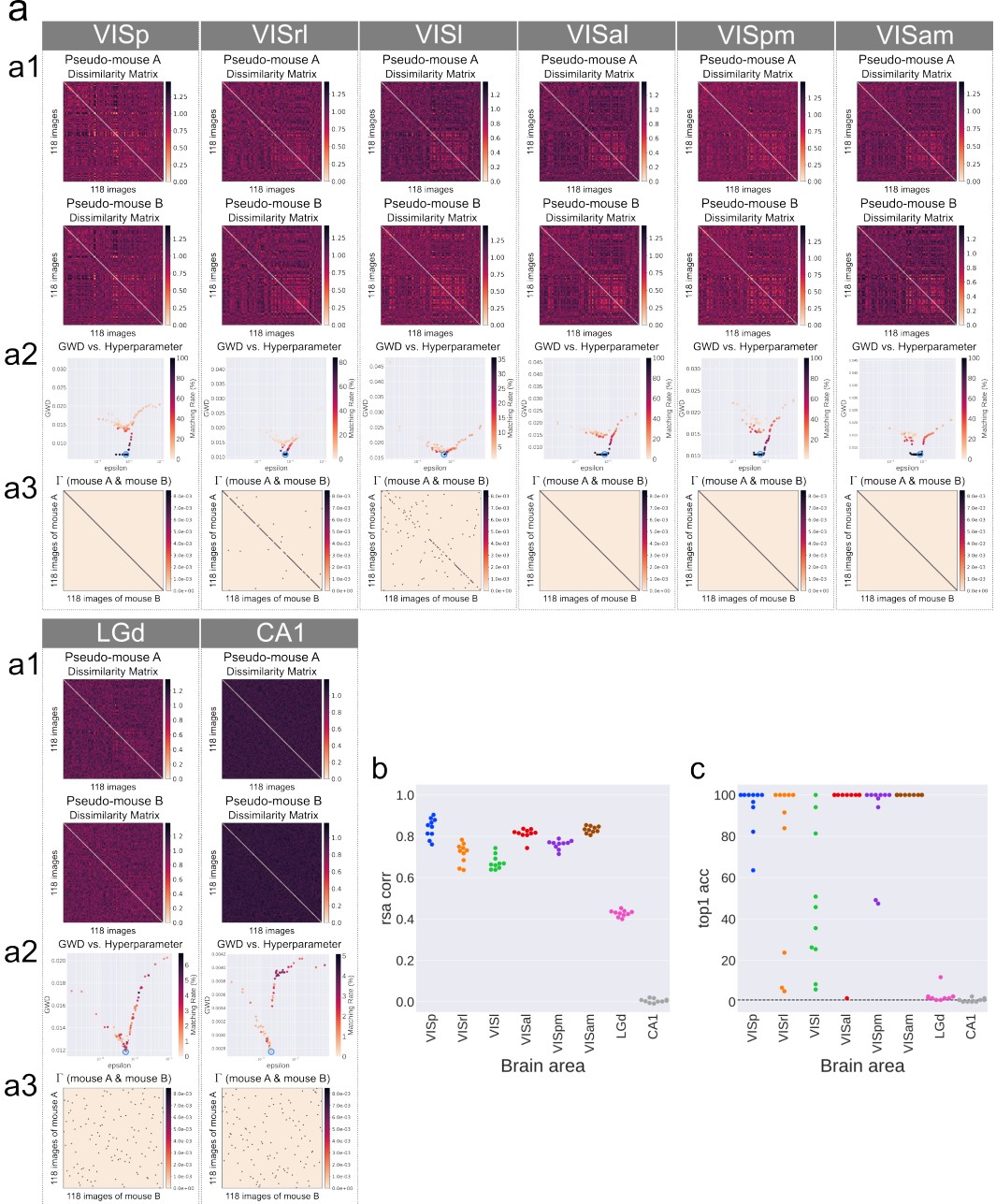

Figure 2: **Unsupervised alignment between the same areas in different pseudo-mice**. (a) The results of GWOT between the dissimilarity matrices of the same areas in different pseudo-mice. (a1) The dissimilarity matrices of a pair of pseudo-mice for the 118 stimuli in each area. (a2) The relationship between GWD (objective of GWOT) and the hyperparameter $\epsilon$. (a3) The optimal transportation plan $\Gamma^*$ between the dissimilarity matrices of a pair of pseudo-mice. This matrix corresponds to the point encircled in blue in Fig. 2a2. (b) The correlation coefficient of RSA between the dissimilarity matrices of a pair of pseudo-mice across 10 trials. (c) The top 1 matching rate of the unsupervised alignment between the dissimilarity matrices of a pair of pseudo-mice across 10 trials.

The optimal transportation plan $\Gamma^*$ obtained through GWOT exhibited distinct characteristics between the visual cortical areas and other areas. Examples of the optimal transportation plan $\Gamma^*$ for specific pairs of pseudo-mice are shown in Fig. 2a3. In most areas of the visual cortex, the optimal transportation plans were close to diagonal matrices, i.e., most of the diagonal elements tend to have higher values than the off-diagonal elements. This means the same stimuli are matched with each other across the different pseudo-mice. In contrast, in LGd, the optimal transportation plan is not

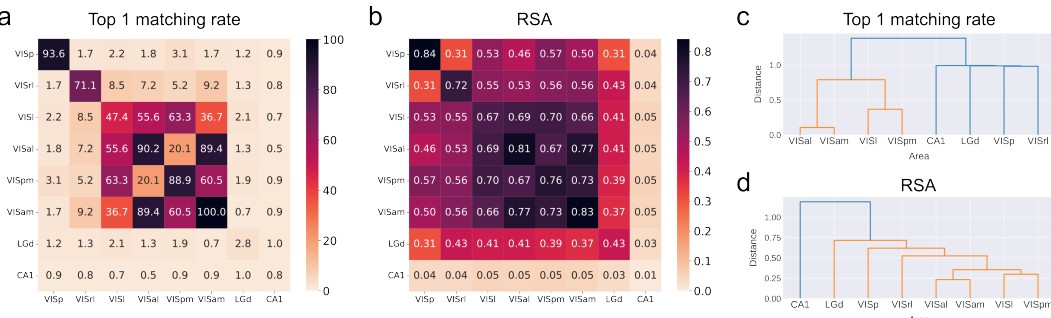

Figure 3: **Unsupervised alignment between the different areas in different pseudo-mice**. (a) The average top 1 matching rate of the unsupervised alignment for each pair of brain areas. (b) The average correlation coefficient of the RSA for each pair of brain areas. (c) Hierarchical clustering of brain areas based on the average top 1 matching rate. Here, the distance between areas is defined as $(100 - \text{top 1 matching rate})/100$ and Ward's method is employed as the clustering criterion. (d) Hierarchical clustering of brain areas based on the average correlation coefficient. The distance between areas is defined as $(1 - \text{correlation})$.

diagonal but rather randomly mapped different stimuli over the pseudo-mice. The failure of unsupervised alignment in LGd is not fully expected because LGd exhibited a moderate degree of structural similarity as indicated in Fig. 2b though the correlations are lower than the visual cortical areas. In comparison, in CA1, the optimal transportation plan also showed random matching but this result is expected because there are no common structures observed at the level of correlation (Fig. 2b). Taken together, these results suggest that the representational similarity structures within the same visual cortical area are highly consistent across animals, while they are notably less consistent in the thalamus and there is no common structure in CA1.

We performed the unsupervised alignment analysis 10 times with different random groupings of animals. The average top 1 matching rates indicated that visual cortical areas generally share unsupervisedly alignable representational structures, while other areas do not (Fig. 2c). The visual cortical areas showed the following average top 1 matching rates across 10 trials: 93.6% for VISp, 71.1% for VISrl, 47.4% for VISl, 90.2% for VISal, 88.9% for VISpm, and 100.0% for VISam, all of which significantly exceeded the chance level of 0.85%. On the other hand, subcortical areas showed the average top 1 matching rates of 2.8% for LGd and 0.8% for CA1, values nearly equivalent to the chance level.

Finally, we compared the results obtained by the unsupervised alignment based on GWOT with the traditional supervised alignment (RSA) to assess the consistency or difference between the results. Upon calculating the average correlations between dissimilarity matrices using the conventional RSA framework, we observed consistent characteristics overall between GWOT (unsupervised, Fig. 2c) and RSA (supervised, Fig. 2b), yet there were subtle differences. For instance, while the dissimilarity matrices of LGd exhibited moderate correlations of 0.4, their top 1 matching rates are almost 0%. Additionally, the matching rates for VISl showed high variability, while the correlation values were much less variable and concentrated around 0.7. The important observation here is that high correlations do not necessarily mean the high matching rates in unsupervised alignment.

**Unsupervised alignment between the different brain areas**   To investigate the extent to which representational similarity structures can be aligned across different brain areas, we also performed an unsupervised alignment of dissimilarity matrices between different areas of two pseudo-mice. Similar to the same area experiment, we varied the random grouping of the animals 10 times and calculated the average top 1 match rate. It is worth mentioning that a single grouping of mice can yield two pairs of different areas, such as VISp of pseudo-mouse A and VISrl of pseudo-mouse B, as well as VISrl of pseudo-mouse A and VISp of pseudo-mouse B. Therefore, the resulting value represents the average of the top 1 match rates for 20 pairs.

Analysis of the average top 1 matching rate has provided insights into the commonality of representational similarity structures across different areas (Fig. 3a), with the primary findings being that: 1) the neural representations of VISp and VISrl exhibit a unique similarity structure distinct from those of other areas, and 2) the representational similarity structures of the higher-order visual cortical areas are similar to each other. First, the average top 1 matching rates between VISp and other areas are approximately 0–3%, which is remarkably low compared to the value of 93.6% observed within

VISp itself, and close to the chance level of 0.85%. This suggests that the neural representations of VISp possess a unique similarity structure, differing not only from those in the thalamus and hippocampus but also from those in other visual cortical areas. A similar tendency was observed for VISrl, although not as pronounced as for VISp. Second, within VISl, VISal, VISpm, and VISam, there are pairs of areas that exhibit high top 1 matching rates exceeding 50%. This result reveals that there can be commonalities in representational similarity structures in the higher-order visual cortical areas, even among distinct areas. Moreover, similar insights were inferred from the results of the conventional RSA framework (Fig. 3b), though the relationships between areas were observed more clearly through unsupervised alignment.

To further interpret the relationships between different areas as quantified by unsupervised alignment, we conducted hierarchical clustering of the areas based on the average top 1 matching rate (Fig. 3c). This hierarchical structure closely matches our earlier findings. Notably, VISp and VISrl form independent clusters, distinct from other visual cortical areas. Meanwhile, within the higher-order visual cortex, smaller distances between VISl and VISpm, as well as between VISal and VISam, indicate the formation of 2 sub-clusters.

Finally, we compare the hierarchical clustering results obtained by unsupervised alignment based on GWOT with those obtained by supervised alignment (RSA) to assess the consistency or difference between the results. Using average correlation coefficients from the conventional RSA framework for hierarchical clustering, we observed a different cluster structure compared to the top 1 matching rate result (Fig. 3d). In the higher-order visual cortex, the formation of 2 sub-clusters obtained from the RSA was consistent with the results observed in the top 1 matching rate of unsupervised alignment. However, in contrast to the results of unsupervised alignment, where other areas formed independent single clusters, we observed a gradual increase in distance from the higher-order visual cortex in the order of VISrl, VISp, and LGd. This order strongly reflects the functional hierarchy of the mouse visual system (Siegle et al., 2021). Thus, the comparison suggests that the inter-regional relationships based on unsupervised alignment via GWOT are not simply predictable from RSA correlations or hierarchical proximity.

## 2.2 UNSUPERVISED ALIGNMENT OF HUMAN NEURAL REPRESENTATIONS

Next, we investigated whether the neural representations of natural scenes in the human brain could be aligned across participants using the Neural Scenes Dataset (NSD) (Allen et al., 2022) (see Methods A.2 for the data description). As in the previous section, we first conducted comparisons within the same areas to investigate whether similarity structures in the same brain areas can be aligned across individuals. Next, we performed cross-areal comparisons to investigate the extent to which representational similarity structures can be aligned across different brain areas.

**Representational similarity structures in each brain area** First, we estimated the representational similarity structures of visual stimuli from 515 natural scenes for several visual areas (V1, V2, V3, pVTC, and aVTC) in each participants group. In Fig. 4a1, we show a specific example of pairs of the dissimilarity matrices of the 515 stimuli in each area in two participants groups. We computed the mean dissimilarity matrix of the 515 stimuli for each area in each participant group, where the dissimilarity between stimuli is quantified by the correlation distance $(1 - \rho)$ between the vectors of neural responses to the stimuli (see Methods A.3 for details). As seen in Fig. 4a1, the appearance of the similarity structure in each region appears similar between the two groups. To quantitatively demonstrate the degree of similarity, we calculated the correlations between the two matrices. Fig. 4b shows the correlations for ten randomly divided group pairs, showing high correlations in all regions. Notably, the correlations are particularly strong in higher-order regions (pVTC and aVTC) than in lower-order regions (V1, V2, and V3).

**Unsupervised alignment between the same brain areas** We next investigated whether the neural representations of the same brain regions could be aligned between participants groups. To this end, we performed unsupervised alignment between the mean dissimilarity matrices of the same brain regions in different participants groups. We show the results of one of the 10 groupings in Fig. 4a, and all the results of the 10 different samples in Figs. 4b–4e. We first performed GWOT between the two dissimilarity matrices for each brain region (Fig. 4a1). We performed a total of 200 optimization iterations on different $\epsilon$ values. The points in Fig. 4a2 correspond to the local

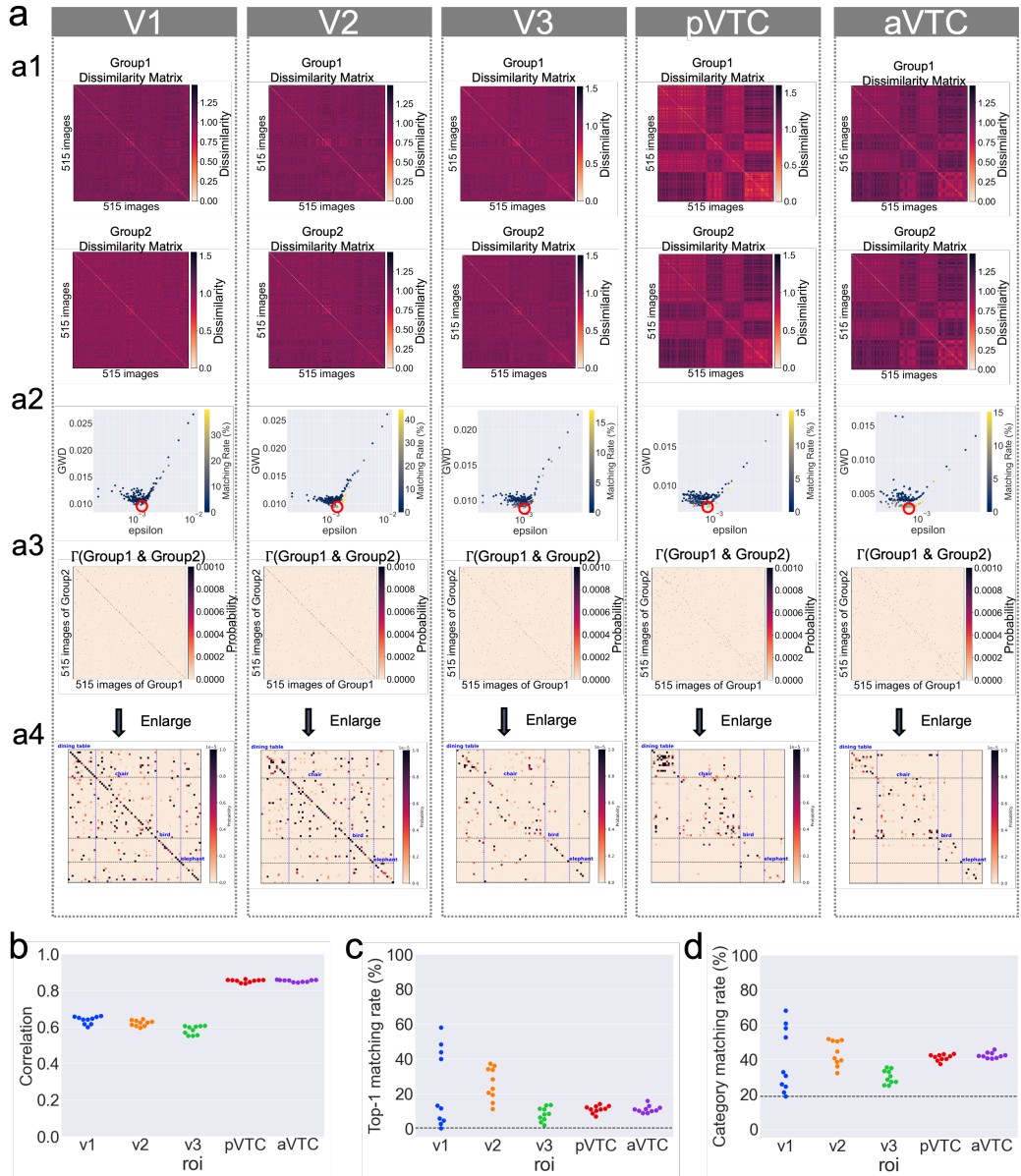

Figure 4: **Unsupervised alignment between the same brain areas in different participants groups**. (a) The results of GWOT between the dissimilarity matrices of the same brain areas in different participants groups. (a1) The dissimilarity matrices of 515 stimuli of Group1 and Group2. (a2) Relationship betwen GWD and matching rate. Color represents top 1 matching rates. (a3) The optimal transportation plan $\Gamma^*$ between the dissimilarity matrices of Group1 and Group2. (a4) Enlarged view of the some coarse categories of the optimal transportation plan $\Gamma^*$. Blue dotted lines indicated the boundaries of the coarse categories. (b) The top 1 matching rate of the unsupervised alignment for 10 random pairs of participant groups. (c) The top 1 category matching rate of the unsupervised alignment for 10 random pairs of participant groups. (d) The RSA correlation coefficient between the dissimilarity matrices of the same brain areas in different participants groups.

minimum found in each iteration of the optimization performed on different $\epsilon$ values. We selected the local minimum with the lowest GWD as the optimal solution (shown in the red circle in Fig. 4a2). After the optimization, we obtained the optimal transportation plan $\Gamma^*$ between the two dissimilarity matrices (Fig. 4a3).

The optimal transportation plan $\Gamma^*$ obtained through GWOT indicates the structural commonality between the representational similarity structures of two different participant groups. As shown in Fig. 4a3, in many cases, the diagonal elements in $\Gamma^*$ have high values, indicating that many of the

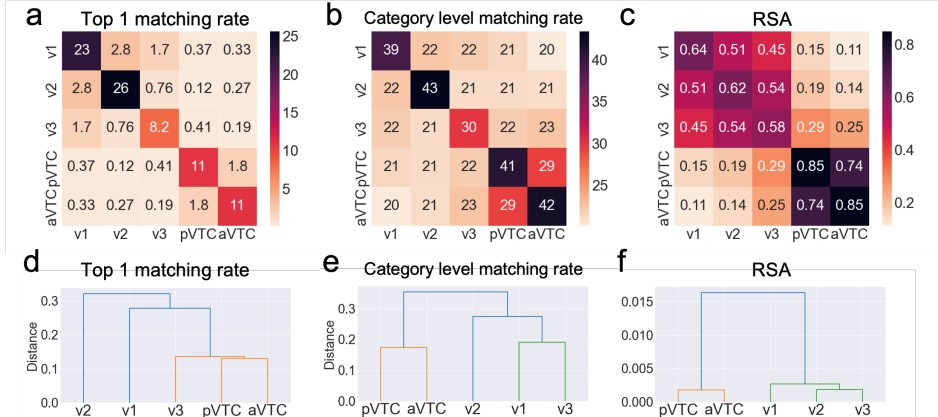

Figure 5: **Unsupervised alignment between different brain areas**. (a) The average top 1 matching rate of the unsupervised alignment for each pair of brain areas. (b) The average category matching rate of the unsupervised alignment for each pair of brain areas. (c) The average RSA correlation coefficient between the dissimilarity matrices of different brain areas. (d) The dendrogram of the hierarchical clustering based on the average matching rate of the unsupervised alignment. (e) The dendrogram of the hierarchical clustering based on the average category matching rate of the unsupervised alignment. (f) The dendrogram of the hierarchical clustering based on the average RSA correlation coefficient between the dissimilarity matrices of different brain areas.

stimuli in one group correspond to the same stimuli in the other group with high probability. Fig. 4a4 is an enlarged view of the optimal transportation plan $\Gamma^*$ shown in Fig. 4a3. As particularly indicated by pVTC and aVTC in Fig. 4a4, even in the case of mismatches, matching errors tend to occur within the same coarse category such as dining table, chair, bird and elephant.

By performing the same analysis with 10 different random groupings of the participants, we obtained the top 1 matching rate of the 10 random pairs (Fig. 4c). The average of the top 1 matching rate over 10 random groupings is 23% for V1, 26% for V2, 8.2% for V3, 11% for pVTC, and 11% for aVTC. We also calculated the top 1 category matching rate of the 10 random samples (Fig. 4d). The average of the top 1 category matching rate over 10 random samples is 39% for V1, 43% for V2, 30% for V3, 41% for pVTC, and 42% for aVTC. These values are significantly higher than the chance level (0.2% for the top 1 matching rate and 18.7% for the top 1 category matching rate). These results suggest that the representational structures in the same brain regions are similar enough to allow for matching through unsupervised alignment.

**Unsupervised alignment between the different brain areas** Next, we investigated whether the similarity structures of neural representations of natural scenes in different brain regions could be aligned across participants. To this end, we performed unsupervised alignment between the mean dissimilarity matrices of different brain regions in different participants groups.

For each case, we performed GWOT between the two dissimilarity matrices. We then calculated the matching rate of the unsupervised alignment, and averaged the matching rate over 10 random groupings of participants. In Fig. 5a, we show the average top 1 matching rate of the unsupervised alignment between the different brain regions in different participants groups. The average of the top 1 matching rate over 10 random samples is 2.8% for V1-V2, 1.7% for V1-V3, 0.37% for V1-pVTC, 0.33% for V1-aVTC, 0.76% for V2-V3, 0.12% for V2-pVTC, 0.27% for V2-aVTC, 0.41% for V3-pVTC, 0.19% for V3-aVTC, 1.8% for pVTC-aVTC. The matching rates between the different brain areas were slightly higher than the chance level but overall much lower than the matching rate between the same brain regions. At this level of matching rate, we cannot observe a clear relationship between the different regions.

On the other hand, the category-level matching of the unsupervised alignment revealed a slightly clearer relationship between regions. For category-level matching rates, only pVTC and aVTC showed slightly higher values, reaching 29% (Fig. 5b). In other comparisons, however, the matching rates were close to the chance level (18.7%). Overall, the unsupervised alignment indicated the closeness of pVTC and aVTC, but other trends were not so clearly observed.

Similar but much clearer tendency was observed in the framework of RSA. In Fig. 5c, we show the RSA correlation coefficient between the dissimilarity matrices of different brain regions. The

correlation coefficients showed high values in the comparisons within V1, V2, V3, and within pVTC and aVTC, whereas they were low in the comparisons across these divisions. This trend is consistent with what was observed in unsupervised alignment, but it has become clearer and more discernible.

To visually elucidate the relationships between the different brain regions, we applied hierarchical clustering to both the matching rate and the RSA correlation (Figs. 5d–5f). We used $(100-\text{matching rate})/100$ as the inter-region distance for the matching rate, and $1-\text{RSA correlation}$ as the inter-region distance for the RSA correlation. We employed Ward's method as the criterion. In all three cases, pVTC and aVTC were classified into the closest clusters. Furthermore, for the clustering based on the category-level matching rate and RSA, we can see that V1, V2, V3 form a different cluster than pVTC and aVTC.

## 3 DISCUSSION

In this study, we proposed a novel framework to compare neural representational similarity structures without using stimulus identities, but purely based on the internal relations of the representational structures, using an unsupervised alignment method based on GWOT. A fundamental difference between this framework and the conventional supervised framework is that this framework attempts to find the optimal correspondences between neural representations of stimuli and to assess whether neural representations of the same stimuli are actually mapped across different individuals, challenging the implicit assumption behind the conventional supervised comparison. We consider that if two similarity structures are aligned at the one-to-one item level with the unsupervised alignment, this provides evidence for a stronger and finer level of structural correspondences beyond simple correlations.

In our analysis of the mouse Neuropixels dataset, we found that the similarity structures of neural representations can be well aligned in the same visual cortical areas (VISp, VISrl, VISl, VISal, VISpm, VISam) but not in the thalamus (LGd), suggesting regional differences in the degree of commonality of representational structures across individuals. Given that LGd is located at the bottom of the mouse visual system hierarchy after the retina, and that LGd has strong feedforward projections into VISp (Harris et al., 2019; Siegle et al., 2021), it is not clear why LGd lacks alignable structures while VISp (or higher-order visual areas) have consistent alignable structures across animals. Future work combining experimental and computational neuroscience would be needed to elucidate what kind of representational transformation from the thalamus to the visual cortex makes representations more consistent.

In addition, using unsupervised alignment, we obtained inter-regional structural relationships in the mouse visual system that are consistent with conventional knowledge, but still show some distinctive features that are not fully expected by conventional analysis. In particular, we found that neural representations in VISp and VISrl could not be aligned with other visual cortical areas closely related in the hierarchy, suggesting that these areas have idiosyncratic structures. In contrast, neural representations were well aligned within higher-order visual cortical areas (VISl, VISal, VISpm, VISam), and the dorsal-like regions VISl and VISpm or the ventral-like regions VISal and VISam (Bakhtiari et al., 2021) are more aligned with each other, as shown in the hierarchical clustering analysis (Fig. 3c). Although these are reasonable results given the known functional similarities of these brain areas, we consider it still meaningful to show that these higher-order visual areas share enough common structures to allow unsupervised alignment.

In analyzing the human fMRI data, we also found that neural representations of the same brain regions can be unsupervisedly aligned across individuals to some extent, although the matching rate is lower than in the mouse data. This is probably due to the difference between the types of recordings, i.e. invasive recordings from Neuropixels and non-invasive recordings from fMRI, and the number of subjects used for averaging the similarity matrices, 16 for the mouse data and 4 for the fMRI data. Given that fMRI typically contains a lot of noise in the data in general, it may be significant that unsupervised alignment was successful even with an average of only 4 participants, indicating the high quality of this dataset and the strong commonalities between individuals. Although less clear than in the case of the mouse data, we were also able to assess the proximity of some regions using unsupervised alignment between different regions. In particular, in the pVTC-aVTC, it was possible to assess the proximity of similarity structures by considering category-level matching. Future re-

search is expected to expand the analysis scope of the brain regions and comprehensively investigate their relationships.

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

## A  METHODS

### A.1  COMPARISON OF NEURAL REPRESENTATIONS

#### A.1.1  REPRESENTATIONAL SIMILARITY STRUCTURES

To compare neural representations across individuals, we use similarity structures of neural representations, which we call representational similarity structures. The neural representation of stimuli is characterized by the activity patterns of individual neurons, with each stimulus represented as a point in a space of neuron dimensions. Suppose we want to compare the neural representations between individuals A and B (Fig. 1). Naively, one might think to directly compare the vectors in two domains, but this approach is not feasible because the correspondence of neuron groups characterizing stimuli cannot usually be estimated between individuals (Fig. 1a). However, by focusing on the similarity structures, it becomes possible to compare neural representations beyond individuals. By measuring the distance between all pairs of representations within the same space, we obtain representational dissimilarity matrices (RDM). The dissimilarity matrix represents the distances between representations of each stimulus in a brain, forming a matrix of the size of the stimulus set by the stimulus set. In this study, we use the dissimilarity matrix to compare similarity structures across individuals. There are two methods for comparing the two similarity structures: supervised alignment and unsupervised alignment.

#### A.1.2  SUPERVISED ALIGNMENT

A simple approach is to compare structures assuming a correspondence between stimulus labels, namely, comparing them in a supervised manner (Fig. 1b). The representative method is Representational Similarity Analysis (RSA) (Kriegeskorte et al., 2008a; Kriegeskorte & Kievit, 2013). RSA is a method to calculate how similar two dissimilarity matrices are. Specifically, it is computed as follows: Given two dissimilarity matrices, $D$ and $D'$, first, the lower triangular part of each matrix is extracted, forming vectors $d$ and $d'$ respectively. Then, the Spearman correlation coefficient between $d$ and $d'$ is calculated. This correlation coefficient serves as an indicator of the similarity between the two dissimilarity matrices.

#### A.1.3  UNSUPERVISED ALIGNMENT

Another approach is to compare structures without assuming a correspondence between stimulus labels, namely, comparing them in an unsupervised manner (Fig. 1c). Generally, unsupervised alignment is a methodology for finding the optimal mappings between items in different domains when the correspondences between the items are completely unknown or not entirely given. As a promising approach to unsupervised alignment, the Gromov-Wasserstein optimal transport (GWOT) method (Mémoli, 2011) has been applied with great success in various fields: for example, matching of 3D objects(Mémoli, 2011), translation of vocabularies in different languages(Alvarez-Melis & Jaakkola, 2018; Alaux et al., 2018), and matching of single cells in single-cell multi-omics data(Demetci et al., 2022); and in neuroscience, alignment of different individual brains in fMRI data(Thual et al., 2022) and comparison of color similarity structures between different individuals (Kawakita et al., 2023a;b). We used the GWOT method to compare the neural representations of natural scenes in different individuals.

**Gromov-Wasserstein optimal transport**  Gromov-Wasserstein optimal transport (Mémoli, 2011) is an unsupervised alignment technique that finds the correspondence between two point clouds in different domains based only on internal distances within each domain. In our case, each point represents a neural representation of natural scenes. Mathematically, the goal of the Gromov-Wasserstein optimal transport problem is to find the optimal transportation plan $\Gamma$ between the two point clouds in different domains, given the internal dissimilarity matrices $D$ and $D'$ within each domain (Fig. 6). The transportation cost, i.e., the objective function, considered in GWOT is given by

$$\min_{\Gamma} \sum_{i,j,k,l} (D_{ij} - D'_{kl})^2 \Gamma_{ik} \Gamma_{jl}. \tag{1}$$

Note that a transportation plan $\Gamma$ must satisfy the following constraints: $\sum_j \Gamma_{ij} = p_i$, $\sum_i \Gamma_{ij} = q_j$, $\sum_{ij} \Gamma_{ij} = 1$ and $\Gamma_{ij} \geq 0$, where $p$ and $q$ are the source and target distributions of resources for the transportation problem, respectively, whose sum is 1. Under this constraint, the matrix $\Gamma$ is

Figure 6: **Unsupervised alignment of neural representation using Gromov-Wasserstein optimal transport**. $D_{ij}$ represents the distance between the $i$th point and the $j$th point in the representational similarity structure A and $D_{kl}$ represents the distance between the $k$th point and the $l$th point in the other representational similarity structure B. $\Gamma_{ik}$ represents the amount of transportation from the $i$th point in A and the $k$th point in B. The optimal correspondence $\Gamma$ is estimated by minimizing the Gromov-Wasserstein Distance (GWD).

considered as a joint probability distribution with the marginal distributions being $p$ and $q$. As for the distributions $p$ and $q$, we set $p$ and $q$ to be the uniform distributions. Each entry $\Gamma_{ij}$ describes how much of the resources on the $i$-th point in the source domain should be transported onto the $j$-th point in the target domain. The entries of the normalized row $\frac{1}{p_i}\Gamma_{ij}$ can be interpreted as the probabilities that the $i$-th point in the source domain corresponds to the $j$-th point in the target domain.

**Evaluation of unsupervised alignment** To evaluate the extent to which the points (stimulus images) were matched with their correct pairs by the obtained optimal transportation plan, we used the measure termed "correct matching rate", defined as follows. For an element $i$, the following function checks if the transportation amount $\Gamma_{ii}$ to its counterpart in the other domain is the maximum among the amounts to any other elements:

$$\text{Match}(i) = \begin{cases} 1, & \text{if } \arg\max_j(\Gamma_{ij}) = i \\ 0, & \text{otherwise.} \end{cases} \tag{2}$$

By using this value, the matching rate is then defined as the percentage of points that are matched with their correct pairs:

$$\text{Matching rate} = \frac{\sum_{i=1}^n \text{Match}(i)}{n}. \tag{3}$$

In the human fMRI dataset, the stimulus images are classified into 80 categories. To evaluate the extent to which the points were matched with the points in the same category, which are not necessarily the exact correct pairs, we used the measure "category-level matching rate", defined as follows. When $i$ belongs to a category $C$, the following function checks whether the element $j$ that receives the maximum amount of transportation from the element $i$ is in the same category $C$ as $i$:

$$\text{Category-level match}(i) = \begin{cases} 1, & \text{if } \arg\max_j(\Gamma_{ij}) \in C \\ 0, & \text{otherwise.} \end{cases} \tag{4}$$

The category-level matching rate is then defined as the percentage of indices that are matched with any indices within the same category:

$$\text{Category-level matching rate} = \frac{\sum_{i=1}^n \text{Category-level match}(i)}{n}. \tag{5}$$

## A.2 DATA

To investigate whether the neural representations of natural scenes in different individuals can be aligned, we used two datasets: the mouse Neuropixels dataset from Allen Brain Observatory Visual Coding (Siegle et al., 2021) and the human fMRI dataset from the Neural Scenes Dataset (NSD) (Allen et al., 2022). All data are available from `http://brain-map.org/explore/circuits` for the mice Neuropixels and from `http://naturalscenesdataset.org` for the human fMRI.

### A.2.1 Mouse Neuropixels dataset: Allen Brain Observatory Visual Coding

We utilized the Allen Brain Observatory Visual Coding - Neuropixels dataset (Siegle et al., 2021). This dataset contains extracellular electrophysiology recordings of neural activities in various brain regions (visual cortex, hippocampus, thalamus, and midbrain) of mice, obtained by Neuropixels probes during the presentation of various types of visual stimuli. Among them, we focused on natural scenes stimuli in this study. Although we also analyzed natural movie 1 and natural movie 3 and obtained qualitatively similar results, we present only the analysis of natural scenes here due to the limitation of the space. The natural scenes stimuli consist of 118 black-and-white images of natural scenes, each presented 50 times in random order. There were a total of 32 mice for the experiment of natural scenes stimuli, and all of them were used in the analysis.

### A.2.2 Human fMRI dataset: Neural Scenes Dataset

The Natural Scenes Dataset (NSD) (Allen et al., 2022) consists of high-resolution (1.8-mm) whole-brain 7T functional magnetic resonance imaging (fMRI) of 8 human participants who each viewed 9,000-10,000 color natural scenes, which are different scenes among participants. The set of natural scenes includes the special subset of 515 images that were commonly presented across participants. We used the recordings corresponding to the shared 515 images for the analysis. In the experiment, each image was presented three times to a given participant. To control the cognitive state and encourage deep processing of the images, participants were instructed to perform a continuous recognition task in which they reported whether the current image had been presented at any previous point in the experiment.

### A.3 Representational similarity structures

### A.3.1 Mouse Neuropixels dataset

**Extracting neural representations** In the analysis of the mouse dataset, trial-averaged vectors of normalized spike counts were used as neural representations for the natural scenes stimuli. Initially, for each neuron in each mouse, we counted the number of spikes during the image presentation (250 ms), followed by standard normalization. Given that each image was presented 50 times, we averaged the normalized spike counts over these 50 trials to obtain the neural representations for each image for every mouse.

**Brain regions** Among the recorded brain areas in the dataset, we analyzed the following 8 areas in the visual cortex and subcortical regions: 6 areas of the visual cortex (VISp: primary visual area, VISrl: rostrolateral visual area, VISl: lateromedial visual area, VISal: anterolateral visual area, VISpm: posteromedial visual area: VISam: anteromedial visual area), 1 area of the thalamus (LGd: dorsal part of the lateral geniculate nucleus), and 1 area of the hippocampus (CA1: cornu ammonis 1). The 6 areas in the visual cortex and LGd are components of the visual system, while CA1 is a part of the memory system. Not all brain areas were measured in each mouse, and the number of neurons used for our analysis ranged from 20 to 200 per mouse per region, with an average of approximately 60.

**Group-averaged representational similarity structures** To obtain a statistically reliable estimate of representational similarity structures, we group-averaged the neural activity of multiple mice, considered as a "pseudo-mouse". To construct a pair of pseudo-mice, we randomly divided the total of 32 mice provided in the dataset into two non-overlapping groups of 16 mice each. We then concatenated the neural representations within each group along the neurons. Therefore, the neural representations for each pseudo-mouse are represented by vectors of 400 to 1,200 neurons, varying by the brain area. Using these aggregated neural representations, we estimated the group-averaged representational similarity structures for the 118 natural scene stimuli in each area for each pseudo-mouse. The dissimilarity between two stimuli was quantified using cosine distance, based on the trial-averaged normalized spike counts.

### A.3.2 Human fMRI dataset

We followed the data preprocessing procedure of the original study (Allen et al., 2022).

**Extracting neural representations**   We extracted the single neural response vector for each 515 images across 8 participants. By applying the Generalized Linear Model (GLM), we obtained the single-trial betas, which are the estimates of the fMRI response amplitude of each voxel to each trial conducted. The single-trial betas were averaged across the three repetitions of each image. Betas for each surface vertex were z-scored within each scan session, concatenated across sessions and averaged across repeated trials for each distinct image. The resulting single-trial betas were used as the neural representation of each 515 images.

**Brain regions**   Following the previous study(Allen et al., 2022), we defined a set of brain regions (V1, V2, V3, pVTC and aVTC) on the fsaverage surface. This was done by mapping the manually defined V1, V2 and V3 from each participant to fsaverage, averaging across participants and using the result to guide the definition of group-level brain regions. We also defined a posterior and anterior division of the ventral temporal cortex (pVTC and aVTC, respectively) based on anatomical criteria. For each participant, we extracted betas for vertices within each brain region (concatenating across hemispheres). The extracted betas were used as the neural representations of the 515 images for each brain region.

**Group-averaged representational similarity structure**   We randomly divided 8 participants into two non-overlapping groups which consisted of 4 participants, and estimated the group-averaged representational similarity structure for each group. We repeated this random dividing procedure 10 times. To estimate the group-averaged representational similarity structure, we first estimated the representational structure for each participant, and then averaged these in the group. The procedure was as follows. For each brain region in each participant, we first extracted a single vector of neural responses to each 515 stimulus. We next computed the dissimilarity matrix of the 515 stimuli for each brain region in each participant, where the dissimilarity between stimuli is quantified by the correlation distance $(1 - \rho)$ between the vectors of neural responses to the stimuli. Finally, for each group, we calculated the mean dissimilarity matrix of each brain region by computing the average values for each entry of matrices.

