# OpenReview forum: "Unsupervised alignment reveals structural commonalities and differences in neural representations of natural scenes across individuals and brain areas"
_ICLR.cc/2024/Workshop/Re-Align — ICLR 2024 Workshop Re-Align Poster_

### Official Review · Reviewer_gADK · 2024-02-23
**Motivation?**

**Rating:** 1
**Fit:** 3
**Confidence:** 3

**Workshop Review:**

This work proposes an alternative method for comparing neural representations that can be compared to RSA. It is based on optimal transport and does not assume a 1-1 correspondence between stimulus representations across brains.

I don't understand why you would question this assumption. Is there any realistic or relevant example where we might have reason to believe that stimulus A neural representation in brain 1 is more similar to stimulus B representation in brain 2? I cannot think of any scenario where this would be a reasonable assumption.

A few random comments:

penultimate line of abstract: -'a'

what do you mean by 'supervised'?

Fig1b, right typo in title

page 2 second paragraph, I guess I don't understand this

page 3 top, what is your null model when checking 'whether two brain regions can be aligned'? I.e., how do you determine whether you have been successful, e.g., more or less so than RSA?!?

What are 'pseudo-mice'?

Define GWD

page 3, bottom, if the diagonal gives the best match, doesn't that question the assumption that optimal stimulus matches shift across brains?

Fig2 is it fair to say that c is the same final outcome for GWOT as b is for RSA? If so, I am confused. It seems like the selectivity of LGd is lost, but nothing is won, i.e., RSA tells me more about what is going on

page 5 first sentence: -'are'

page 5 second paragraph, what does this mean?

page 5 3rd paragraph, it seems like GWOT is more variable? I guess this is something I am wondering anyways. RSA is a linear method, GWOT depends on some non-convex optimization. The latter is only preferable if it really buys is more insights, which does not seem to be the case (see above)

page 6 top, are those insights from GWOT reliable? How can we test their significance



It would be nice to see comparisons with other neural metrics:

https://proceedings.neurips.cc/paper_files/paper/2021/file/252a3dbaeb32e7690242ad3b556e626b-Paper.pdf

https://openreview.net/pdf?id=LhV3Ex8fky

**Reason For Not Giving Higher Score:**

I don't understand the motivation for the method

**Reason For Not Giving Lower Score:**

n/a

**Reviewer Domain:**

neuroscience

---

### Official Review · Reviewer_gWPR · 2024-02-23
**Well motivated study**

**Rating:** 2
**Fit:** 3
**Confidence:** 3

**Workshop Review:**

This work employs a relatively novel alignment metric, Gromov-Wasserstein Optimal Transport (GWOT), to compare neural representations' structures across different brain regions or among the same region in different individuals. Unlike commonly used similarity measures like Representational Similarity Analysis (RSA), GWOT offers a more flexible approach by relaxing the stimulus correspondence assumption, making it a better fit for such comparisons.

Although the study is well-conceived and the application of this alignment metric to analyze widely recognized datasets (such as those from Allen and Allen et al.) is insightful, it appears that these analyses have yet to yield significant new findings.

**Reason For Not Giving Higher Score:**

At this point in this research, the novelty of applying this method and the significance of findings derived from using this alignment metric is not clear

**Reason For Not Giving Lower Score:**

Using GWOT for the widely used datasets is informative for the community.

**Reviewer Domain:**

neuroscience

---

### Official Review · Reviewer_Fgow · 2024-02-23
**Extremely fitting, well-written, well-executed**

**Rating:** 3
**Fit:** 3
**Confidence:** 2

**Workshop Review:**

I think this work would be of particular interest to the community. It revolves around extending RSA to account for possible permutations in the samples (so not assuming a given correspondence in the stimuli). This is accomplished using Gromov-Wasserstein Optimal Transport, therefore relying on well-established tools.

My background is in machine learning, but I had no problem following the work therefore, I can also rate the writing clarity high.

The experimental setup seems robust, with repeated runs and different groupings to support the results.

Regarding the novelty, I'm unfamiliar with existing neuroscience literature on the topic, so it's difficult to assess in this regard.

**Reason For Not Giving Higher Score:**

N/A

**Reason For Not Giving Lower Score:**

I think it's already part of the review section.

**Reviewer Domain:**

machine learning

---

### Decision · Program_Chairs · 2024-03-02

Accept (Poster)